# Structural basis of guanine nucleotide exchange for Rab11 by SH3BP5

Sakurako Goto-Ito[1,2], Nobukatsu Morooka[3], Atsushi Yamagata[1,2,4], Yusuke Sato[1,2,4], Ken Sato[3] , Shuya Fukai[1,2,4]

**The Rab GTPase family is a major regulator of membrane traffic in eukaryotic cells. The Rab11 subfamily plays important roles in specific trafficking events such as exocytosis, endosomal recycling, and cytokinesis. SH3BP5 and SH3BP5-like (SH3BP5L) proteins have recently been found to serve as guanine nucleotide exchange factors (GEF) for Rab11. Here, we report the crystal structures of the SH3BP5 GEF domain alone and its complex with Rab11a. SH3BP5 exhibits a V-shaped structure comprising two coiled coils. The coiled coil composed of α1, and α4 is solely responsible for the Rab11a binding and GEF activity. SH3BP5 pulls out and deforms switch I of Rab11a so as to facilitate the GDP release from Rab11a. SH3BP5 interacts with the N-terminal region, switch I, interswitch, and switch II of Rab11a. SH3BP5 and SH3BP5L localize to Rab11-positive recycling endosomes and show GEF activity for all of the Rab11 family but not for Rab14. Fluorescence-based GEF assays combined with site-directed mutagenesis reveal the essential interactions between SH3BP5 and Rab11 family proteins for the GEF reaction on recycling endosomes.**

## Introduction

The Rab family is the largest small GTPase family of the Ras superfamily. More than 60 Rab proteins have been identified in human. They are major regulators of membrane traffic (1). Each Rab protein shows distinctive localizations in plasma membrane and/or organelles such as endosomes, Golgi, and lysosome. Rab is reversibly anchored to membranes by the posttranslational geranylgeranylation of the C-terminal cysteine residues. The membrane-anchored Rab proteins serve as identity markers for membrane compartments and control the downstream events in membrane traffic.

Like other small GTPases, Rab functions as a molecular switch by shuttling between the GTP-bound, active state and the GDP-bound, inactive state (2) and has three functional motifs: switch I, switch II,

and P loop. Switch I and switch II adopt different conformations, depending on the nucleotide-bound state. The P loop interacts with phosphate moieties of the bound guanine nucleotide. In addition, the region between switch I and switch II is termed the interswitch region, which is often functionally important. Rab guanine nucleotide exchange factors (GEFs) stimulate the conversion of the GDP-bound, inactive Rab to the GTP-bound, active Rab. Mechanistically, RabGEFs accelerate the removal of the bound GDP, and the resultant nucleotide-free Rab eventually captures GTP, which is more abundant than GDP in the cytosol. Conversely, GTPase activating proteins (GAPs) enhance the intrinsic GTPase activity of Rab, accelerating the conversion of Rab-GTP to Rab-GDP. In short, the functional state of Rab is bidirectionally controlled by GEFs and GAPs. Most RabGAPs contain Tre-2/Bub2/Cdc16 (TBC) domains as the catalytic domain, whereas the catalytic mechanism of RabGEFs is known for its variety, as represented by diverged GEF domain structures such as DENN, Vps9, Sec2, TRAPP, and Mon1-Ccz1 (3, 4, 5, 6, 7, 8, 9, 10).

Mammals have three Rab11 isoforms: Rab11a, Rab11b, and Rab11c (also known as Rab25). Rab11a is ubiquitous, whereas Rab11b and Rab11c are expressed in specific tissues. They all localize in the *trans*-Golgi network, post-Golgi vesicles, and recycling endosomes and play central roles in exocytotic and recycling pathways (11). For instance, Rab11 regulates the arrangement of cell surface proteins, including membrane receptors and cell adhesion proteins on the plasma membrane. Rab11 directly interacts with the exocyst complex, which tethers secretory vesicles to the plasma membrane (12). Also, Rab11 is included in various motor protein complexes for transport of membrane vesicles (13, 14). Because of the fundamental role of Rab11 in vesicular transport, many pathogens target Rab11 for infection (15). Pathogens use or block the host Rab11 pathway for their invasions. Some viruses exploit the host Rab11 pathway for their exit from cells.

Despite the functional importance of Rab11, mammalian Rab11 GEF had not been clearly identified for a long time. Although a DENN family protein named Crag was previously shown to deprive GDP from Rab11, the activity for Rab11 was much weaker than that for Rab10 (16). Three years ago, *C. elegans* REI-1/2 and its mammalian

[1]Institute for Quantitative Biosciences, The University of Tokyo, Tokyo, Japan   [2]Synchrotron Radiation Research Organization, The University of Tokyo, Tokyo, Japan   [3]Laboratory of Molecular Traffic, Institute for Molecular and Cellular Regulation, Gunma University, Maebashi, Japan   [4]Department of Computational Biology and Medical Sciences, Graduate School of Frontier Sciences, The University of Tokyo, Chiba, Japan

Correspondence: fukai@iam.u-tokyo.ac.jp; sato-ken@gunma-u.ac.jp

homologue SH3-binding protein 5 (SH3BP5) were identified as Rab11 GEFs (17), and fairly recently, Parcas, the drosophila homologue of SH3BP5, was also shown to possess the GEF activity for Rab11 (18). SH3BP5 is a conserved Rab11 GEF among species. Of note, SH3BP5 was predicted to comprise a novel RabGEF family, as it has no sequence similarity to other known RabGEFs. Therefore, we set out to obtain structural insights into this novel RabGEF. During the preparation of this article, the crystal structure of SH3BP5–Rab11a complex was reported by Burke and his colleagues (19). Their structure is similar to our structure, supporting the mechanism that we clarify in this article. Here, we present the crystal structures of human apo-SH3BP5 and Rab11a-bound SH3BP5. The structures demonstrate that SH3BP5 comprises a novel RabGEF family. The structure-based mutagenesis, together with in vitro GEF assays, reveals a GEF mechanism of SH3BP5 specific for the Rab11 family on recycling endosomes. We also show that SH3BP5 and SH3BP5L localize to Rab11-positive recycling endosomes and have GEF activity for all of the Rab11 family but not for Rab14.

## Results

### Structure of SH3BP5

Human SH3BP5 is composed of the N-terminal acidic region (residues 1–43), central helical region (residues 44–262), and C-terminal region (residues 263–455) (Fig 1A). Because computational secondary structure prediction (20) assigned the N- and C-terminal regions as mostly disordered regions, we selected the central helical region for crystallography. However, the C-terminal portion of the purified central region (residues 41–266) was partially degraded, as confirmed by SDS–PAGE and N-terminal sequencing (Fig S1A). We found that the R260A/R261A/R262A triple mutation reduced the degradation and obtained diffraction-quality crystals (Fig S1B). This construct of SH3BP5 (41–266; R260A/R261A/R262A) is hereafter referred to as SH3BP5-RA. In addition, the M167A mutation was introduced to improve the solubility of the selenomethionine (SeMet)-labeled SH3BP5-RA (SeMet SH3BP5-RA/M167A), which was used for phase determination by the single-wavelength anomalous diffraction method. Finally, we determined the crystal structures of SeMet SH3BP5-RA/M167A in two different crystal forms ($I4_1$ and $P4_1$) and native SH3BP5-RA at 3.35, 3.6, and 3.8 Å resolutions, respectively (Table 1). The asymmetric units of the $I4_1$ and $P4_1$ crystals contain one and two SH3BP5 molecules, respectively. Therefore, we obtained five different apo-SH3BP5 structures in total (i.e., one from $I4_1$ SeMet SH3BP5-RA/M167A, two from $P4_1$ SeMet SH3BP5-RA/M167A, and two from $P4_1$ native SH3BP5-RA). Although these five apo-SH3BP5 structures exhibit a similar V-shaped conformation, the superposition of these five apo-SH3BP5 structures using α1 as the reference shows some flexibility of the conformation (Fig 1B).

SH3BP5 has four α-helices (α1–α4), which form two coiled coils (Fig 1C): one is formed by α1 and α4, whereas the other is formed by α2 and α3. The hinge region of the V shape contains staggered hydrophobic interactions so as to maintain the V shape (Fig 1D). Leu84, Leu87, and Val88 on α1, Val95 on α2, and Pro210, Tyr211, and Phe212 on α4 form a hydrophobic core. Another hydrophobic core is composed of Tyr101 and Trp102 on α2, Met194 and Leu197 on α3, and

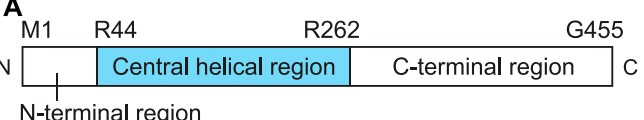

**A**

M1    R44              R262                    G455

N [ | Central helical region | C-terminal region | ] C

N-terminal region

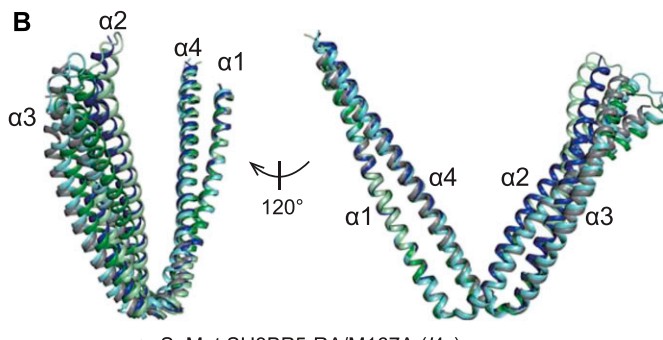

**B**

◆ SeMet SH3BP5-RA/M167A ($I4_1$)
◆ SeMet SH3BP5-RA/M167A ($P4_1$)
◆ Native SH3BP5-RA ($P4_1$)

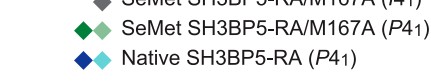

**C**

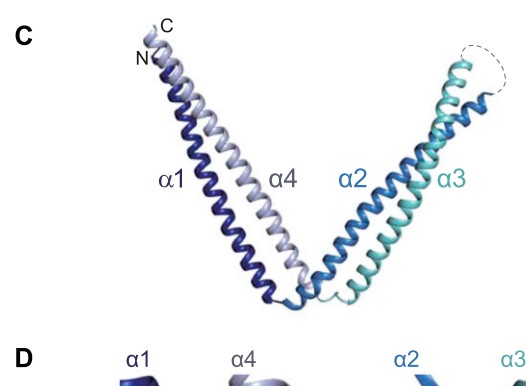

**D**

**Figure 1. Structure of apo-SH3BP5.**
**(A)** Domain organization of SH3BP5. **(B)** Superposition of apo-SH3BP5 structures from different crystal forms. **(C)** Structure of apo-SH3BP5. α1, α2, α3, and α4 are colored in dark blue, blue, cyan, and pale blue, respectively. **(D)** Hydrophobic interactions in the hinge regions.

Ile205 on α4. The stacking interaction between Trp102 on α2 and Phe212 on α4 combines these two hydrophobic cores to reinforce the hinge region. The residues involved in these hydrophobic interactions are conserved or replaced by functionally equivalent residues among SH3BP5 and SH3BP5-like proteins from representative metazoa (Fig S2). The V-shaped structure is likely to be a common feature of SH3BP5 and SH3BP5-like proteins.

### Structure of SH3BP5–Rab11a complex

To elucidate the GEF mechanism of SH3BP5, we also determined the crystal structure of SH3BP5 in the complex with Rab11a at 3.8 Å

**Table 1.  Data collection and refinement statistics.**

| Molecule name | SeMet SH3BP5-RA/M167A | SeMet SH3BP5-RA/M167A | Native SH3BP5-RA | SH3BP5–Rab11a |
|---|---|---|---|---|
| PDB ID | 6IXE | 6IXF | 6IXG | 6IXV |
| Data collection | | | | |
| Beamline | SPring-8 BL41XU | SPring-8 BL41XU | SPring-8 BL41XU | SPring-8 BL41XU |
| Space group | $I4_1$ | $P4_1$ | $P4_1$ | $I222$ |
| Cell constants | | | | |
| $a, b, c$ (Å) | 79.3, 79.3, 108.1 | 78.3, 78.3, 93.0 | 80.1, 80.1, 95.0 | 117.8, 199.1, 303.9 |
| $\alpha, \beta, \gamma$ (°) | 90.0, 90.0, 90.0 | 90.0, 90.0, 90.0 | 90.0, 90.0, 90.0 | 90.0, 90.0, 90.0 |
| Resolution | 50–3.35 (3.41–3.35) | 50–3.6 (3.66–3.6) | 50–3.8 (3.87–3.8) | 50–3.8 (3.87–3.8) |
| $R_{sym}$ | 0.158 (0.357) | 0.181 (0.704) | 0.187 (0.707) | 0.157 (1.546) |
| $I/\sigma I$ | 30.0 (2.52) | 14.9 (1.04) | 14.9 (1.42) | 41.1 (1.32) |
| Redundancy | 8.2 (3.2) | 8.2 (3.9) | 9.1 (5.1) | 32.6 (16.4) |
| Completeness (%) | 96.9 (89.0) | 97.5 (92.3) | 94.4 (80.1) | 100.0 (100.0) |
| Refinement | | | | |
| Resolution (Å) | 38.9–3.35 | 47.5–3.6 | 40–3.8 | 49.8–3.8 |
| No. reflections | 4,689 | 6,398 | 5,578 | 35,564 |
| $R_{work}/R_{free}$ | 0.269/0.314 | 0.234/0.261 | 0.242/0.278 | 0.229/0.259 |
| No. atoms | | | | |
| Protein | 1,737 | 3,613 | 3,577 | 12,750 |
| Ligand/ion | 8 | 0 | 0 | 15 |
| $B$-factors (Å²) | | | | |
| Protein | 122.7 | 119.3 | 140.2 | 224.9 |
| Ligand/ion | 127.3 | — | — | 270.9 |
| Rmsds | | | | |
| Bond lengths (Å) | 0.003 | 0.003 | 0.003 | 0.001 |
| Bond angles (°) | 0.658 | 0.604 | 0.678 | 0.461 |
| Ramachandran plot | | | | |
| Favored (%) | 99.0 | 97.5 | 99.5 | 97.0 |
| Allowed (%) | 1.0 | 2.5 | 0.5 | 3.0 |
| Outliers (%) | 0.0 | 0.0 | 0.0 | 0.0 |

resolution (Fig 2A and Table 1). Wild-type SH3BP5 (10–276) was used for crystallization of the complex because SH3BP5-RA tended to crystallize in the apo form. The asymmetric unit contains four complexes, where SH3BP5 interacts with Rab11a in a similar manner (Fig S3A and B). The electron density corresponding to the N- and C-terminal regions of SH3BP5 (residues 10–41 and 268–276, respectively) and the N-terminal four residues of Rab11a were invisible, suggesting the structural disorder of these regions. We hereafter describe one of the four complexes (corresponding to chains A [SH3BP5] and E [Rab11a] in PDB 6IXV), whose electron density was clearer than those of the other three complexes, unless otherwise noted.

In the SH3BP5–Rab11a complex, both switch regions (switch I and switch II) of Rab11a interact with the coiled coil comprising α1 and α4 of SH3BP5. As compared with the GTP- or GDP-bound Rab11a structures, the switch I region is pulled out by SH3BP5 so as to

facilitate the release of the bound guanine nucleotide and Mg²⁺ (21) (Fig 2). The switch II region adopts a helical structure, which is also observed in the inactive, GDP-bound state of Rab11a (Fig 2A and B). In contrast, in the active, GTP-bound state, α-helix in switch II is partly distorted to enable the main-chain NH group of Gly69 to interact with the γ-phosphate group of GTP (Fig 2C) (21). In summary, SH3BP5 opens the guanine nucleotide-binding pocket by capturing both switch regions. On the other hand, the P loop region, which binds to β- and γ-phosphate groups, does not contact SH3BP5 and remains folded as in the GDP- and GTP-bound structures.

The SH3BP5–Rab11a complex structure suggested that the α1/α4 coiled coil is solely responsible for the interaction with Rab11a. The α1/α4 coiled coil indeed retains the GEF activity (Fig S3C). When superposing the four Rab11a-bound SH3BP5 structures in the asymmetric unit using Rab11a as the reference, α1 and α4 helices in all these structures are also superposed well (Fig S3B). On the other

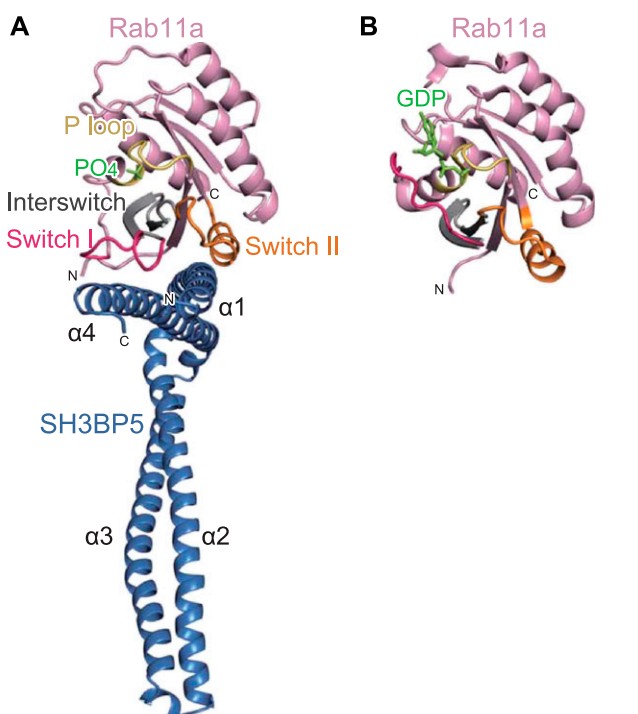

**Figure 2. Structure of the SH3BP5–Rab11a complex.**
**(A)** Overall view of the SH3BP5–Rab11a complex. Rab11a is colored in pink, except that switch I, switch II, interswitch, and P loop are colored in red, orange, grey, and yellow, respectively. SH3BP5 is colored in blue. The phosphate ion bound to the P loop is shown as light green sticks. **(B)** Structure of the GDP-bound Rab11a. The coloring scheme is the same as that in (A). GDP is shown as light green sticks. **(C)** Structure of the GTPγS-bound Rab11a. The coloring scheme is the same as that in (A). GTPγS and Mg²⁺ are shown as light green sticks and a purple sphere, respectively.

hand, the α2/α3 coiled coil shows some flexibility, even in the Rab11a-bound state. A noted structural difference was observed in the edge of the α2/α3 coiled coil, which is disordered or fixed by crystal packing in two drastically different conformations (Fig S3B). This flexibility of the edge of the α2/α3 coiled coil, which is located far from the GEF catalytic site, might have some switching functions independent of the GEF reaction, such as binding to other proteins or membrane lipids.

**Interactions between SH3BP5 and Rab11a**

SH3BP5 interacts with the N-terminal region, interswitch, switch I, and switch II of Rab11a in the SH3BP5–Rab11a complex. The contribution of each interaction site to the GEF activity was examined by fluorescence-based in vitro assay of site-directed mutants. The observed pseudo-first-order rate constant ($k_{obs}$) for each mutant was calculated from the reaction curve. For this assay, we used SH3BP5-RA as a standard SH3BP5 protein because SH3BP5-RA could be handled much more easily than SH3BP5 (41–266, wild-type) and retains the GEF activity. Rab11a (1–173) and Rab11a (1–211) were used for the assays of SH3BP5 and Rab11a mutants, respectively. SH3BP5-RA showed similar GEF activities toward both Rab11a (1–173) and Rab11a (1–211), although Rab11a (1–173) seemed more susceptible to EDTA than Rab11a (1–211) in terms of the rate of guanine nucleotide exchange (Fig S4A). A rapidly decreasing phase was observed in the beginning of the curves without GEF. Mant-GDP used in this study was a mixture of 2′-Mant-GDP and 3′-Mant-GDP. 2′-Mant-3′-deoxy-GDP reportedly has 10 times faster dissociation rate than 3′-Mant-2′-deoxy-GDP (22). The rapidly decreasing phase might reflect the faster release of 2′-Mant-GDP.

The N-terminal region of Rab11a interacts with α1 of SH3BP5. Asn60 of SH3BP5 forms two hydrogen bonds with the Leu11 main chain of Rab11a (Fig 3A). The N60A mutant of SH3BP5 had no GEF activity, indicating the importance of these hydrogen bonds (Figs 3B and S4B). Although Thr64 of SH3BP5 forms a hydrogen bond with Tyr10 of Rab11a, the T64A mutant of SH3BP5 showed higher GEF activity than the wild type (Figs 3A, B, and S4C). Consistently, the Y10A mutation of Rab11a also showed higher GEF activity (Figs 3B and S4C).

The interswitch region of Rab11a interacts with both α1 and α4 of SH3BP5. Phe48 and Trp65 of Rab11a hydrophobically interact with Leu49, Leu52, Tyr243, and Leu247 of SH3BP5 (Fig 3C). This hydrophobic interaction is critically important because the alanine mutations of these residues drastically decreased the GEF activity (Figs 3D and S4B). Next to this hydrophobic interaction, Gln63 of Rab11a forms a hydrogen bond with Tyr243 of SH3BP5. However, this hydrogen bond is dispensable for the GEF activity because the Q63A mutation of Rab11a showed little effect on the GEF activity (Figs 3C, D and S4C). Therefore, the decrease in GEF activity of SH3BP5 Y243A mutant depends mainly on the loss of hydrophobic interaction.

The switch I region of Rab11a is pulled out and interacts with the C-terminal region of SH3BP5 α4 (Fig 4A). Although the electron density of the switch I region was not well resolved, we traced the main chain. To confirm the residue assignment and further assess the contribution of each residue in the switch I region to the GEF activity, we prepared the L38A, E39A, S40A, K41A, S42A, T43A, and I44A mutants of Rab11a and subjected them to the GEF assay. The L38A and E39A mutations reduced the activity to ~50% of the wild-type activity, and the S40A, K41A, and S42A mutations reduced it to ~25% of the wild-type activity (Figs 4A, B, and S4C). On the other hand, the

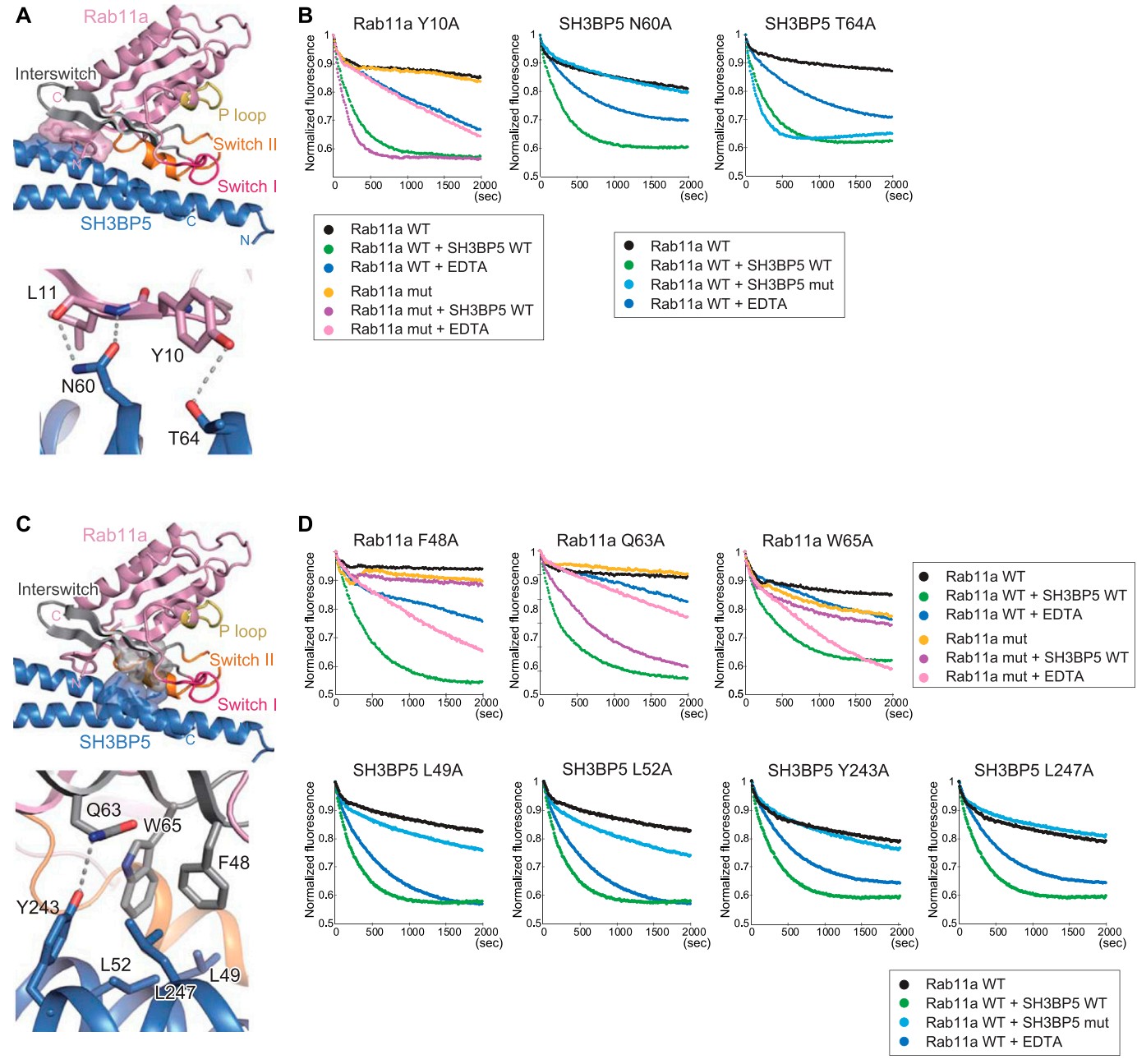

**Figure 3. Interactions of SH3BP5 with the N-terminal region and interswitch of Rab11a.**
**(A)** Interactions of SH3BP5 with the N-terminal region of Rab11a. The coloring scheme is the same as that in Fig 2A. The interacting residues of SH3BP5 and Rab11a are shown as sticks (with translucent molecular surfaces in the top panel). The SH3BP5 residues interacting with Rab11a are shown as blue sticks with translucent molecular surface. The close-up view of the interactions is also shown in the bottom. Hydrogen bonds are indicated as dotted lines. **(B)** Guanine nucleotide exchange assays for Rab11a and SH3BP5 mutants that were designed to become defective in the interaction between SH3BP5 and the N-terminal region of Rab11a. The time courses of the Mant-GDP fluorescence are plotted for the indicated samples. One representative plot from three or more independent experiments is shown for each sample. **(C)** Interactions of SH3BP5 with the interswitch region of Rab11a. The presentation scheme is the same as that in (A). **(D)** Guanine nucleotide exchange assays for Rab11a and SH3BP5 mutants that were designed to become defective in the interaction between SH3BP5 and the interswitch region of Rab11a. The time courses of the Mant-GDP fluorescence are plotted for the indicated samples. One representative plot from three or more independent experiments is shown for each sample.

T43A and I44A mutations eliminated the activity (Figs 4B and S4C). These results are consistent with our modeling of the switch I region and indicated that Ser40–Ile44, especially Thr43 and Ile44, are important for the interaction between SH3BP5 and the switch I region of Rab11a. However, it should be noted that the side chain of

Thr43 coordinates $Mg^{2+}$ in the nucleotide-bound Rab11a. The T43A mutation might affect the nucleotide binding, although Mant-GDP could be loaded to the Rab11a T43A mutant as efficiently as to wild type. In the present SH3BP5–Rab11a structure, Ser254 and His258 of SH3BP5 form hydrogen bonds with the Glu39 main chain and Thr43

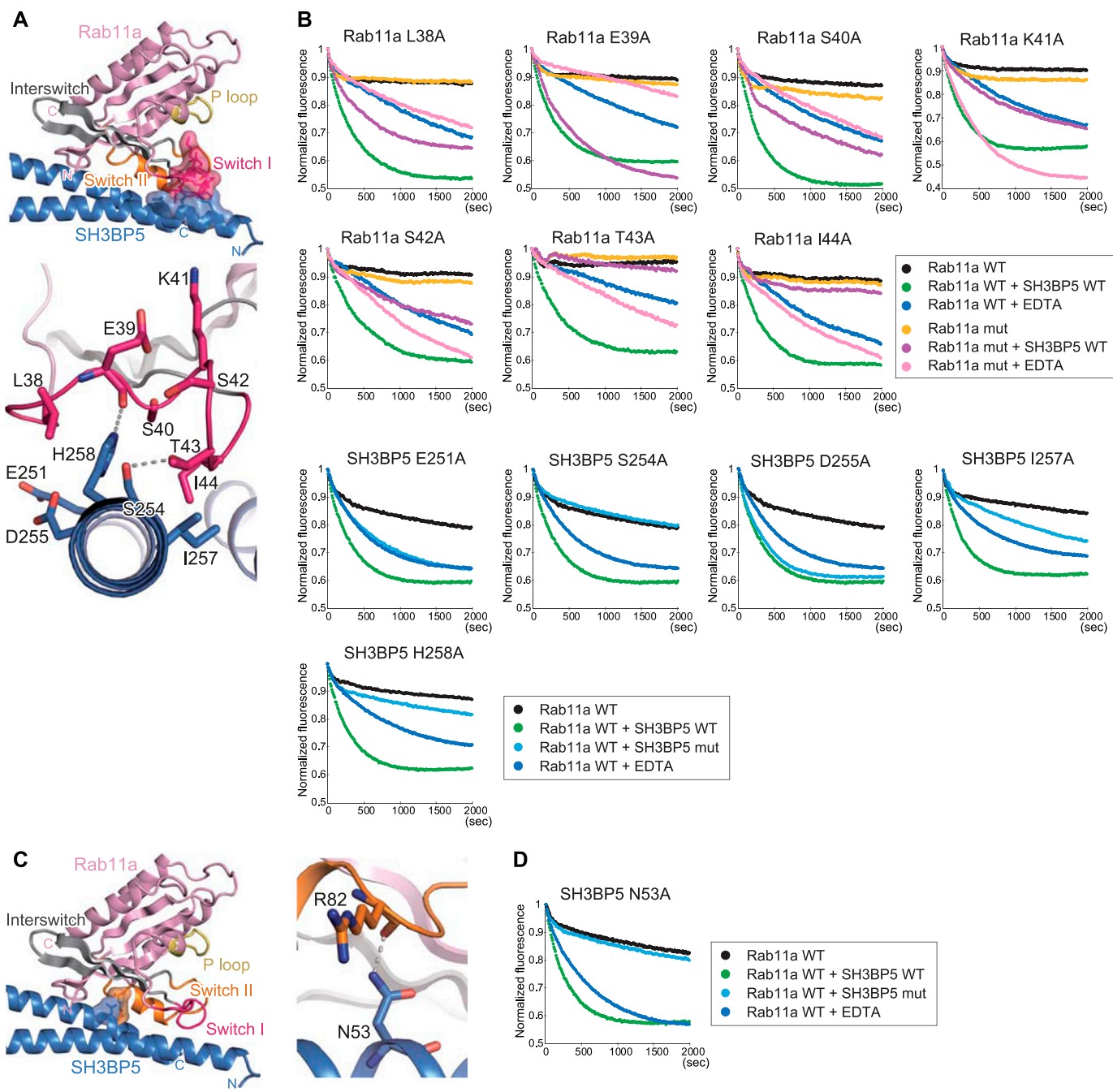

**Figure 4.  Interactions of SH3BP5 with the switch regions of Rab11a.**
**(A)** Interactions of SH3BP5 with the switch I region of Rab11a. The presentation scheme is the same as that in Fig 2A. **(B)** Guanine nucleotide exchange assays for Rab11a and SH3BP5 mutants that were designed to become defective in the interaction between SH3BP5 and the switch I region of Rab11a. The time courses of the Mant-GDP fluorescence are plotted for the indicated samples. One representative plot from three or more independent experiments is shown for each sample. **(C)** Interactions of SH3BP5 with the switch II region of Rab11a. The presentation scheme is the same as that in (A). **(D)** Guanine nucleotide exchange assays for an SH3BP5 mutant that was designed to become defective in the interaction between SH3BP5 and the switch II region of Rab11a. The time courses of the Mant-GDP fluorescence are plotted for the indicated samples. One representative plot from three or more independent experiments is shown.

of Rab11a, respectively. Ile257 of SH3BP5 hydrophobically interacts with Ile44 of Rab11a. The functional importance of these interactions was shown by drastic decrease in the GEF activity by the S254A, I257A, and H258A mutations of SH3BP5 (Figs 4A, B, and S4B). On the other hand, the E251A and D255A mutations of SH3BP5, which are

located far from the switch I residues critical for the GEF activity, showed moderate (~50%, relative to wild type) and little effect on the GEF activity, respectively (Figs 4A, B, and S4B).

In addition, the main chain of Arg82 in switch II of Rab11a forms a hydrogen bond with Asn53 of SH3BP5 (Fig 4C). The N53A mutant of

SH3BP5 showed no GEF activity. This hydrogen bond is essential for the nucleotide exchange activity (Figs 4D and S4B).

### SH3BP5 and SHBP5L mainly localize to Rab11-positive recycling endosomes

We finally investigated subcellular localization of SH3BP5 and SH3BP5L in mammalian cells. We transiently expressed a GFP fusion protein of SH3BP5 or SH3BP5L in Hela cells and compared its localization with organelle marker proteins. GFP-SH3BP5 and GFP-SH3BP5L colocalized with endogenous Rab11 but not with an early endosome marker EEA1 or late endosome/lysosome marker Lamp1 (Fig 5A–H), indicating that SH3BP5 and SH3BP5L mainly localize to recycling endosomes at steady state. Although previous studies suggested that SH3BP5 mainly localizes to mitochondria (23), we hardly detected the signal of GFP-SH3BP5 and GFP-SH3BP5L on MitoTracker-stained mitochondria (Fig S5A and B). We further tested if SH3BP5 and SH3BP5L have GEF activity specific for Rab11 family proteins. For this assay, we added liposomes containing nickel-nitrilotriacetic acid (NTA), which immobilizes His$_6$-tagged Rab proteins on the membrane, to the reaction mixture to make SH3BP5 and SH3BP5L show the full GEF activity in vitro, as reported previously (17). Both of SH3BP5 and SH3BP5L showed strong GEF activity toward purified Rab11a, Rab11b, and Rab25 but not toward Rab14, which localizes to recycling endosomes as well (Fig 5I). These results suggest that SH3BP5 and SH3BP5L mainly localize to recycling endosomes and function as specific GEFs for Rab11 family proteins.

## Discussion

In this study, we revealed that the V-shaped SH3BP5 promotes the nucleotide release from Rab11a by inducing a drastic conformational change of the nucleotide-binding pocket of Rab11a. We further identified key residues of human SH3BP5 and Rab11a for the nucleotide exchange reaction. The SH3BP5 residues that are shown to be responsible for the activity (i.e., Leu49, Leu52, Asn53, Asn60, Tyr243, Leu247, Ser254, and His258 in human SH3BP5) are completely conserved in SH3BP5 and SH3BP5-like proteins from representative metazoa (Fig S2). Similarly, the Rab11a residues especially important in the nucleotide exchange reaction (i.e., Thr43, Ile44, Phe48, and Trp65 in human Rab11a) are completely conserved (Fig S6A). Therefore, the GEF mechanism revealed by the present SH3BP5–Rab11a structure is likely to be evolutionarily conserved.

SH3BP5 and SH3BP5-like proteins exhibit selectivity to the Rab11 subfamily. Therefore, we searched for the determinants in Rab11a/b/c for SH3BP5 substrates. Among the essential Rab11a residues for the GEF reaction, Thr43, Phe48, and Trp65 are perfectly conserved in the Rab family, and Ile44 is replaced by hydrophobic residues (Fig S6B). On the other hand, Ser40, Lys41, and Ser42 of Rab11a, whose mutations moderately affected the GEF reaction, are conserved or replaced by functionally equivalent residues in the Rab11 subfamily but not in the Rab family. It has been reported that no GEF activity of SH3BP5 was detected for the S40F or K41P mutant of Rab11a, which resembles Rab3/8/10 or Rab14, respectively (19). Therefore, the

conformation of the switch I loop is likely to be the critical determinant of Rab11 for SH3BP5 substrate as proposed in (19).

The GEF mechanism of SH3BP5 is different from those of other RabGEFs (Fig S7A). Yeast Sec2p and its mammalian counterpart Rabin8 are coiled-coil GEFs, whose substrates are Sec4p and Rab8, respectively (5, 6, 9). Sec2p and Rabin8 adopt extended conformations in contrast to the V shape of SH3BP5. The binding orientation of Sec2p or Rabin8 to Sec4p or Rab8, respectively, is completely different from that of SH3BP5 to Rab11a (Fig S7B). Sec2p and Rabin8 pull out the switch I region by hydrophobic interactions and thereby disable the nucleotide binding. DENND1B, a member of the DENN domain GEF family, opens the nucleotide-binding pocket of Rab35 by displacing the switch I region and deforms the switch II region to avoid the steric clash between Rab35 and DENN1B (8) (Fig S7C). Vps9 is a globular GEF domain composed of α-helices. The Vps9 domain of Rabex5 catalyzes the guanine nucleotide exchange of Rab5a, Rab21, and Rab22a (4) by physically extruding switch I and disrupting the interaction of the P loop with the β-phosphate group of the bound GDP and the coordinated Mg$^{2+}$ (24) (Fig S7D). DrrA is a *Legionella pneumophila* protein and has a GEF activity for the host Rab1a/1b (25). DrrA pulls out switch I, which forms an α-helix upon binding to DrrA (Fig S7E). DrrA also interacts with switch II and P loop residues. TRAPPI is a multiprotein tethering complex composed of seven subunits and functions as a GEF for Ypt1p and Rab1a/Rab1b. TRAPPI facilitates the nucleotide exchange by the concerted action of the switch I rearrangement and the occupation of the nucleotide-binding pocket by a loop in TRAPPI (7) (Fig S7F). The binding mode of Mon1-Ccz1, a heterodimer GEF active for Rab7a/Rab7b and Ypt7p, to Ypt7p is similar to that of TRAPPI to Ypt1p. Mon1-Ccz1 deforms switch I by hydrophobic interactions, which pushes out the bound Mg$^{2+}$ (10) (Fig S7G). Although the detailed mechanisms are different, all RabGEFs of known structures bind to switch I and alter its conformation drastically.

In this study, we demonstrated that the α1/α4 coiled coil of SH3BP5 retains the GEF activity, suggesting that the α2/α3 coiled coil potentially has other functions than the GEF reaction. As mentioned above, the edge of the α2/α3 coiled coil is basically disordered or fixed by crystal packing in three drastically different conformations (Fig S3B) and might play some switching roles independent of the GEF reaction, possibly through the binding to other proteins and/or membrane lipids. Further functional studies at the molecular and cellular levels are needed to understand how the V shape of SH3BP5 and SH3BP5-like proteins is advantageous in the context of the cellular function.

## Materials and Methods

### Protein preparation

For preparation of human SH3BP5, the region corresponding to human SH3BP5 (41–266) was subcloned into pCold I expression vector (Takara) with the N-terminal His$_6$-SUMO tag. To improve the protein yield, the R260A/R261A/R262A triple mutation was introduced into pCold-His$_6$-SUMO-human SH3BP5 (41–266) (SH3BP5-RA). For preparation of the SeMet-labeled SH3BP5, SH3BP5-RA was

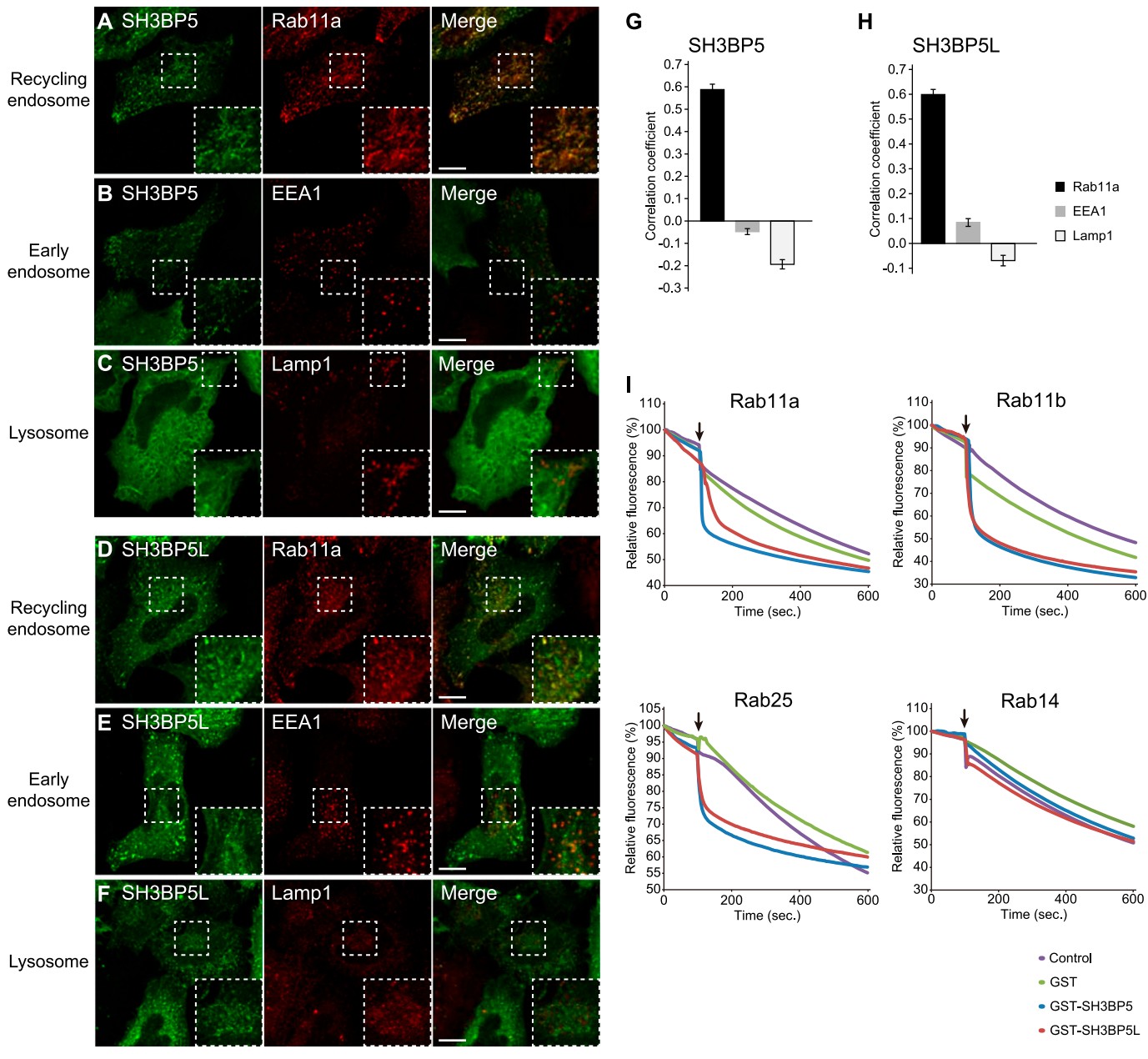

**Figure 5.  SH3BP5 and SH3BP5L localize to recycling endosomes and show GEF activity for Rab11 family proteins.**
**(A–F)** Subcellular localization of EGFP-SH3BP5 and EGFP-SH3BP5L in Hela cells. Hela cells, which transiently expressed EGFP-SH3BP5 or EGFP-SH3BP5L, were immunostained with anti-Rab11a (recycling endosome marker; A and D), EEA1 (early endosome marker; B and E), and Lamp1 (lysosome marker; C and F) as indicated. Scale bars indicate 10 $\mu$m. **(G, H)** Quantitative analysis of colocalization of GFP-SH3BP5 or GFP-SH3BP5L with organelle markers. Pearson's correlation for the colocalization of GFP-SH3BP5 or GFP-SH3BP5L with each organelle marker was calculated by coloc2 analysis performed with Fiji/ImageJ. The graph shows the average colocalization percentage per cell. Error bars represent the SEM ($n$ = 42, 68, and 37 for Rab11, EEA1, and Lamp1 of EGFP-SH3BP5 expressed cells, respectively, and $n$ = 52, 72, and 44 for Rab11, EEA1, and Lamp1 of EGFP-SH3BP5L expressed cells, respectively). **(I)** In vitro GEF assays of SH3BP5 and SH3BP5L for Rab11 family proteins. Mant-GDP release from human Rab11a, Rab11b, Rab25, and Rab14, which are Rab GTPases localized on recycling endosomes, were measured by adding GST-SH3BP5 or GST-SH3BP5L with liposome. Arrows indicate the time points when the hydrolysis-resistant GTP analog was added. One representative plot from two or more independent experiments is shown for each sample.

subcloned into pET28a vector (Merck) with the N-terminal His$_6$-SUMO tag. The M167A mutation was additionally introduced to improve the solubility of the SeMet-labeled SH3BP5-RA. The native His$_6$-SUMO–tagged SH3BP5-RA was expressed in *Escherichia coli* Rosetta (DE3) strain. The cells were cultured at 37°C in LB medium

supplemented with 50 mg/l ampicillin. The protein expression was induced by 0.2 mM IPTG when A$_{600}$ reached 0.6–0.8, and the culture was continued overnight at 15°C. The SeMet-labeled His$_6$-SUMO–tagged SH3BP5-RA/M167A was expressed in *E. coli* B834 strain. The cells were cultured at 37°C in the customized medium equivalent to

LeMaster medium (Code No. 06780; Nacalai tesque) with 50 mg/l SeMet. After the induction by 0.2 mM IPTG, the cells were further grown overnight at 20°C. For both native and SeMet-labeled SH3BP5 proteins, the harvested cells were suspended in 25 mM Tris–Cl buffer (pH 7.4) containing 400 mM NaCl, 2 mM MgCl$_2$, 5 mM imidazole, 5 mM $\beta$-mercaptoethanol, and 0.5% Triton X-100 and were disrupted by sonication. The supernatant was loaded onto an Ni-NTA Superflow (QIAGEN) column pre-equilibrated with the sonication buffer without Triton X-100. After washing with 25 mM Tris–Cl buffer (pH 7.4) containing 200 mM NaCl, 2 mM MgCl$_2$, 50 mM imidazole, and 5 mM $\beta$-mercaptoethanol, the bound protein was eluted with 25 mM Tris–Cl buffer (pH 7.4) containing 200 mM NaCl, 2 mM MgCl$_2$, 250 mM imidazole, and 5 mM $\beta$-mercaptoethanol. The His$_6$-SUMO tag of the eluted protein was cleaved by the SUMO protease Ulp1. The SH3BP5 sample was then subjected to a ResourceS (GE Healthcare) column equilibrated with 25 mM Tris–Cl buffer (pH 7.4) containing 150 mM NaCl, 1 mM MgCl$_2$, and 5 mM $\beta$-mercaptoethanol. The bound protein was eluted by a linear gradient of 150–600 mM NaCl. The eluted SH3BP5 sample was further purified by a Superdex200 10/300 (GE Healthcare) column equilibrated with 25 mM Tris–Cl buffer (pH 7.4) containing 250 mM NaCl and 5 mM $\beta$-mercaptoethanol. For preparation of the $\alpha 1/\alpha 4$ coiled coil of SH3BP5, the DNA regions corresponding to $\alpha 2$ and $\alpha 3$ (residues 95–203) were removed from pET28a-His$_6$-SUMO-SH3BP5-RA plasmid by overlap extension PCR. The His$_6$-SUMO-human SH3BP5 (41–266; Δ95–203) was purified in the same manner as SH3BP5-RA.

For preparation of the SH3BP5–Rab11a complex, human SH3BP5 (10–276) and human Rab11a (1–173) were subcloned into pCold I expression vector (Takara) with the N-terminal GST and His$_6$-SUMO tags, respectively. The GST-tagged SH3BP5 (10–276) and Rab11a (1–173) were expressed in *E. coli* Rosetta (DE3) strain. The His$_6$-SUMO–tagged Rab11a was purified using Ni-NTA in the same manner as the His$_6$-SUMO–tagged SH3BP5-RA. After the Ulp1 cleavage, the Rab11a sample was subjected to a HiLoad 16/600 Superdex200 pg (GE Healthcare) column equilibrated with 25 mM Tris–Cl buffer (pH 7.4) containing 200 mM NaCl, 1 mM MgCl$_2$, and 5 mM $\beta$-mercaptoethanol. The remaining His$_6$-SUMO was removed by reloading the sample to an Ni-NTA Superflow (QIAGEN) column equilibrated with 25 mM Tris–Cl buffer (pH 7.4) containing 200 mM NaCl, 1 mM MgCl$_2$, and 5 mM $\beta$-mercaptoethanol.

## Crystallization

All samples were crystallized by the sitting drop vapor diffusion method at 20°C. The SeMet-labeled SH3BP5-RA/M167A was concentrated to 5.2 g/l and were mixed with an equal volume of the reservoir solution: 0.1 M succinic acid (pH 7.0) and 12% PEG3350. The SeMet-labeled SH3BP5-RA/M167A crystal was cryoprotected by the addition of glycerol (27.5%) or PEG400 (27.5%) to the reservoir solution for the $I4_1$ crystal and $P4_1$ crystal, respectively, and was flash-cooled by liquid nitrogen. The native SH3BP5-RA crystal appeared from a drop containing equal volumes of 4.1 g/l of the SH3BP5-RA–Rab11a (1–173) complex solution and the reservoir solution: 0.4 M Mg(NO$_3$)$_2$, 0.1 M Tris–Cl (pH 8.0), and 22% PEG8000. The crystal was cryoprotected by the addition of glycerol (26.3%) to the reservoir solution and was flash-cooled with liquid nitrogen. The SH3BP5 (10–276)–Rab11a (1–173) complex was concentrated to

6.2 g/l and was mixed with an equal volume of the reservoir solution: 0.4–0.75 M NaCl, 0.1 M NaH$_2$PO$_4$ (pH 6.8), and 15–17% PEG2000. The crystal was cryoprotected by the addition of xylitol (30%) or glucose (25%) to the reservoir solution and was flash-cooled with liquid nitrogen.

## Data collection and structure determination

All diffraction data sets were collected at 100 K at beamline BL41XU of SPring-8 and were processed with HKL2000 (26) and CCP4 program suite (27). The phase for the SeMet-labeled SH3BP5-RA/M167A data set from the $I4_1$ crystal was calculated using the Phenix software (28). The AutoSol (29) wizard found eight selenium sites, and the successive AutoBuild (30) wizard built an initial atomic model containing 270 residues. This initial model was improved by repetitive cycles of manual model building using Coot (31) and refinement using Phenix (28). The structure of the SeMet-labeled SH3BP5-RA/M167A in the $P4_1$ crystal and that of the native SH3BP5-RA mutant were determined by the molecular replacement method using the program Phaser (32). The SeMet-labeled SH3BP5-RA/M167A structure in the $I4_1$ crystal was used as the search model. For structure determination of the SH3BP5 (10–276)–Rab11a (1–173) complex, three data sets from two crystals were merged. The structure of the SH3BP5–Rab11a complex was determined by the molecular replacement method using Molrep (33). The native SH3BP5-RA structure determined in this study and human Rab11a structure (PDB ID: 4CGY) were used as the search models. The initial model was improved by repetitive cycles of manual model building using Coot (31) and refinement using Phenix (28). Data collection and refinement statistics are shown in Table 1. All structural figures were prepared using the program PyMol (Schrödinger, LLC; https://www.schrodinger.com/pymol).

## GEF assay

Rab11a was purified as described above. Mant-GDP (Molecular Probes) was loaded to Rab11a by incubating 3.7 $\mu$M Rab11a and 48 $\mu$M Mant-GDP in 20 mM Hepes-Na buffer (pH 7.5) containing 150 mM NaCl and 5 mM EDTA for 2 h at room temperature. The Mant-GDP loading was stopped by adding 10 mM MgCl$_2$. The excess Mant-GDP was then removed by a PD-10 desalting column (GE Healthcare). The Mant-GDP-bound Rab11a was eluted with 50 mM Tris–Cl buffer (pH 8.0) containing 150 mM NaCl and 0.5 mM MgCl$_2$. The GEF reactions were initiated by mixing the Mant-GDP–bound Rab11a and SH3BP5 in the presence of 200 $\mu$M guanosine 5′-[$\beta,\gamma$-imido]triphosphate (GppNHp). The fluorescent signals were measured by the EnVision multimode plate reader (Perkin Elmer) every 10 s for 2,000 s ($\lambda$ex = 366 nm, $\lambda$em = 443 nm). The reactions were performed at room temperature (20–25°C). For calculation of $k_{obs}$, the time–signal intensity curve was fitted to a single exponential decay: $I(t) = (I_0 - I_\infty) \exp(-k_{obs}) + I_\infty$, where $I(t)$ is the intensity as a function of time. The curve fitting was performed with Kaleidagraph software (HULINKS). These assays were carried out at least thrice for each sample.

For GEF assays of SH3BP5 and SH3BP5L toward Rab11a, Rab11b, Rab25, and Rab14, we used liposomes containing NTA, which immobilizes His$_6$-tagged Rab proteins on the membrane (17). It has

been reported that the addition of NTA-containing liposomes mimics binding of Rab proteins to the membrane, making RabGEFs show their full activity in vitro (17, 34, 35). We purified GST fusion proteins with SH3BP5 and SH3BP5L and each Rab protein, which was tagged with $His_6$-tag at the C terminus. We examined the GEF activity of GST-SH3BP5 (100 pmol) or GST-SH3BP5L (100 pmol) toward each Rab protein (200 pmol) in the presence of liposomes (POPC:1-palmitoyl-2-oleoyl-$sn$-glycero-3-phosphocholine (52.5%) (Avanti Polar Lipids), DOGS-NTA:1, 2-dioleoyl-$sn$-glycero-3-{[$N$(5-smino-1-carboxypentyl)iminodiacetic acid]succinyl}(30%) (Avanti Polar Lipids), PS:L-$\alpha$-phosphatidylserine (10%) (Avanti Polar Lipids), and PI3P:phosphatidylinositol-1-(1,2-dihexadecanoyl)-3-phosphate (7.5%) (Cayman Chemical), as described previously (17). The GEF assays on liposomes were carried out at least twice for each sample.

### Cell culture and transfection

Hela cells were cultured in Dulbecco's modified Eagle's medium (Wako) containing 10% fetal bovine serum (Sigma-Aldrich). Cells were cultured in media supplemented with penicillin–streptomycin–amphotericin B suspension (Wako) and incubated at 37°C under 5% $CO_2$. Hela cells grown on coverslips or in glass bottom dishes to 40 to 60% confluence were transfected with pcDNA3.1 plasmid harboring EGFP-SH3BP5 or EGFP-SH3BP5L (500 ng/ml), respectively. Fugene HD (Promega) in Opti-MEM (Gibco) was used for plasmid transfection. At 4-h post-transfection, the transfection medium was replaced with fresh growth medium. After a 20-h incubation, the cells were collected.

### Immunofluorescence microscopy

Hela cells, which expressed EGFP-SH3BP5 or EGFP-SH3BP5L transiently, were cultured on coverslips and then washed with PBS twice. The cells were fixed with 4% paraformaldehyde in PBS for 15 min and washed three times. Then, we permeabilized cells with 0.05% saponin and incubated them with PBS containing 5% FBS for 30 min for blocking and treated with the following antibodies. We used anti-Rab11 (1:250 dilution; #71-5300; Invitrogen), anti-EEA-1 (1:250 dilution; #610456; BD Biosciences), and anti-Lamp-1 (1:250 dilution; #sc-20011; Santa Cruz Biotechnology) antibodies as the primary antibodies and goat anti-rabbit or anti-mouse Alexa Fluor 594 (1:1,000; Life Technologies Inc.). We acquired images using an FV1200 confocal microscope (Olympus) with a 100× PlanApo oil immersion lens (1.40 numerical aperture; Olympus). For mitochondria staining, we incubated Hela cells, which expressed EGFP-SH3BP5 or EGFP-SH3BP5L transiently in glass bottom dishes, with a MitoTracker Red CMXRos (0.1 nM; M7512; Invitrogen) for 15 min and washed them with PBS twice for observation.

### Data availability

Coordinates and structure factors of SeMet apo-SH3BP5 (SH3BP5-RA/M167A in $I4_1$), SeMet apo-SH3BP5 (SH3BP5-RA/M167A in $P4_1$), native apo-SH3BP5 (SH3BP5-RA in $P4_1$), and the SH3BP5 (10–276)–Rab11a (1–173) complex have been deposited in the Protein Data Bank under accession codes 6IXE, 6IXF, 6IXG, and 6IXV, respectively.

## Supplementary Information

## Acknowledgements

We thank Drs. Aisa Sakaguchi (Osaka University) and Miyuki Sato (Gunma University) for providing biological reagents and suggestions. We thank the beamline staff of BL41XU at SPring-8 (Hyogo, Japan) for technical help during data collection. This work was supported by JSPS KAKENHI (17K15072 to S Goto-Ito and 17K19377 to K Sato), JST CREST (JPMJCR12M5 to S Fukai), and Takeda Science Foundation (to K Sato).

### Author Contributions

S Goto-Ito: formal analysis, funding acquisition, investigation, and writing—original draft.
N Morooka: formal analysis, investigation, and writing—review and editing.
A Yamagata: validation and investigation.
Y Sato: validation and investigation.
K Sato: conceptualization, supervision, funding acquisition, validation, investigation, project administration, and writing—review and editing.
S Fukai: supervision, funding acquisition, validation, investigation, project administration, and writing—review and editing.

### Conflict of Interest Statement

The authors declare that they have no conflict of interest.

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
