## [Reviewer comments · Life Science Alliance]

Life Science Alliance

Structural basis of guanine nucleotide exchange for Rab11 by SH3BP5

Sakurako Goto-Ito, Nobukatsu Morooka, Atsushi Yamagata, Yusuke Sato, Ken Sato, and Shuya Fukai

DOI: <https://doi.org/10.26508/lsa.201900297>

Corresponding author(s): Shuya Fukai, Institute of Molecular and Cellular Biosciences, University of Tokyo and Ken Sato, Gunma University

Review Timeline:

Submission Date:	2019-01-08
Editorial Decision:	2019-01-29
Revision Received:	2019-03-05
Editorial Decision:	2019-03-05
Revision Received:	2019-03-07
Accepted:	2019-03-08

Scientific Editor: Andrea Leibfried

Transaction Report:

January 29, 2019

Re: Life Science Alliance manuscript #LSA-2019-00297-T

Dr. Shuya Fukai
Institute of Molecular and Cellular Biosciences, University of Tokyo
Life Science Division, Synchrotron Radiation Research Organization
1-1-1 Yayoi, Bunkyo-ku
306 Main Building
Tokyo 113-0032
Japan

Dear Dr. Fukai,

Thank you for submitting your manuscript entitled "Structural basis of guanine nucleotide exchange for Rab11 by SH3BP5" to Life Science Alliance. The manuscript was assessed by expert reviewers, whose comments are appended to this letter.

As you will see, the reviewers appreciate that your work confirms and slightly extend recent findings. We would thus be happy to publish a slightly revised version of your manuscript in Life Science Alliance. Both reviewer #2 and #3 provide constructive input on how to further strengthen your manuscript, mainly by text changes. Additionally, please respond to and address reviewer #3 points d) (second paragraph) and e) carefully.

Thank you for this interesting contribution to Life Science Alliance. We are looking forward to receiving your revised manuscript.

Sincerely,

Andrea Leibfried, PhD
Executive Editor

Life Science Alliance
Meyerhofstr. 1
69117 Heidelberg, Germany
t +49 6221 8891 502
e a.leibfried@life-science-alliance.org
www.life-science-alliance.org

- A letter addressing the reviewers' comments point by point.
- An editable version of the final text (.DOC or .DOCX) is needed for copyediting (no PDFs).
- High-resolution figure, supplementary figure and video files uploaded as individual files: See our detailed guidelines for preparing your production-ready images, <http://life-science-alliance.org/authorguide>
- Summary blurb (enter in submission system): A short text summarizing in a single sentence the study (max. 200 characters including spaces). This text is used in conjunction with the titles of papers, hence should be informative and complementary to the title and running title. It should describe the context and significance of the findings for a general readership; it should be written in the present tense and refer to the work in the third person. Author names should not be mentioned.

B. MANUSCRIPT ORGANIZATION AND FORMATTING:

Full guidelines are available on our Instructions for Authors page, <http://life-science-alliance.org/authorguide>

Reviewer #1 (Comments to the Authors (Required)):

The authors have determined by x-ray crystallography a structure of SH3BP5, a Rab11 GEF with a novel fold and activation mechanism, both by itself and in complex with Rab11. Although modest, their resolution is sufficient for them to elucidate the mechanism for activation. Their proposed mechanism is consistent with their thorough mutational/functional analysis. They also determine the basis for Rab11 selectivity.

The results represent a significant advance in the field Rab regulation and are confirmatory/complimentary of independent work published just slightly earlier (in Sept 2018) by the JE Burke group. The authors of the present manuscript have a more extended study in that they have also determined the structure of SH3BP5 alone, although this is not all that informative and does not add much in terms of mechanistic insight, and they present some localization data.

I strongly support publication of this well-done work by LSA.

Reviewer #2 (Comments to the Authors (Required)):

This is a generally well performed study of SH3BP5 as an exchange factor (GEF) for Rab11. The structural results obtained show that the GEF domain of SH3BP5 forms a v-shaped coiled coil structure, but its complex with Rab11 has a completely different overall topology to that of other GEF:GEF complexes that are coiled coil structures (Sec2 or Rabin8 in complex Sec4 or Rab8, respectively). Nevertheless, and even if the authors go to some lengths to point out the supposedly unique characteristics of this particular GTPase:GEF mechanism, it still adheres to the main principles seen (disturbance of Switch I and Switch II structures, displacement of Mg²⁺ ions) in other cases. The difference is in the manner in which these changes are induced. When discussing the mechanisms of other Rab GEFs, the authors emphasize that in one case, the loss of an interaction of the guanine base with a phenylalanine at the start of the Switch I region is important. However, this is probably true of all Ras family GTPase-GEF interactions, including that described here, and should not be emphasized as an individual feature of one mechanism. Looking at the present structure from the coordinates (from pdb code 6djl), it can be seen that Phe36 is far removed from where the guanine base would be if GDP were bound. This can be nicely seen if the structure is superimposed on that of Rab8 in the Rab8:GDP:Rabin8 structure (pdb code 4lhy).

The results in the manuscript agree essentially with those in an earlier paper by Jenkins et al., published on-line in Nat. Comm. On September 14, 2018. This paper is properly referred to in the manuscript. There are no significant differences in the reported results and interpretations. Common to both papers is a rather poorly resolved structure (3.8 Å in this contribution, 3.1 Å in the already published paper). The relatively poor resolution is perhaps the reason for delaying publication, i.e. the authors were probably still trying to improve their crystals, and have now resigned themselves to having to make do with the poor resolution of their structure (even worse than in the published work). Presumably, all interpretations made could be done on the basis of the electron density map from the authors' data, but they did have access to the coordinates of the "competition", at least from September on (pdb code 6djl deposited on 2018-05-25, released on 2018-09-26).

Below are a few minor points:

The fluorescence assay should not be referred to as "our" assay (in the abstract). This is a standard assay that has been used very frequently since the early 90s of the last century, with minor variations. In the present case, I wonder why the authors used GppNHp to displace mantGDP? This is quite expensive, and due to the symmetrical manner in which GEFs function, GDP could just as well used (it is the MantGDP release that is rate limiting, not the GTP or GDP binding, as long as they are present at high concentration). GDP is much cheaper than GppNHp, and GTP itself is even cheaper. Why not use GTP? The hydrolysis reaction is of no consequence. Another point here: the authors used MantGDP from Molecular Probes, which is a mixture of 2 isomers. It is well known that these can have different properties, although that is probably not of

importance for the semi-quantitative analysis performed here. Still, it would be better in general to use 3'-mant-2'-deoxy GDP, which does not have this problem. As far as I can see, the "competition" did use this, but referred to it as MantGDP (should have been mantdGDP). See below for a further point relevant to these arguments.

In the Introduction, second paragraph: Mg²⁺ is not involved in hydrogen bonds (and cannot be).

In Fig. 4a, "no enzyme" should read "no GEF", since GEF is not, strictly speaking, an enzyme. Also, in this figure, and in Fig. 3, why is there a relatively rapid phase at the beginning of the dissociation of MantGDP without added GEF? This could, conceivably, be due to the difference in binding affinities of the 2 isomers alluded to above. In fact, for HRas, Rensland et al. (Biochemistry 30, 11181-11185 (1991); see p. 1183, right hand column) concluded that the 2' isomer bound almost a factor of 10 more weakly than the 3'-isomer, and that this difference was due to an increased off-rate. A test with 3'-mant-2'-deoxyGDP could resolve this question.

Reviewer #3 (Comments to the Authors (Required)):

SH3BP5 is an exchange factor for the small GTPase Rab11 that acts in membrane traffic on recycling endosomes and the trans-Golgi network. Sakurako et al. describe the crystal structure of the GEF domain of SH3BP5 alone and in a complex with Rab11a. The authors find that SH3BP5 folds into a V shape coiled-coil architecture, and its interaction with Rab11a results in a rearrangement of the Rab switch I region. The rearrangement of switch I releases the GDP bound to the GTPase and so facilitates nucleotide exchange.

This structural analysis is of a good quality and convincing, and it is consistent with a structure for SH3BP5 in complex with Rab11 that was published a few months ago (Jenkins et al (2018) Nat Comms 9, 3772). In addition, the authors of this manuscript extend their work beyond that of Jenkins et al, by performing an extensive analysis of different mutant forms of both proteins, Rab11 and SH3BP5, in GEF assays. Overall this is a valuable addition to the field and it both confirms and extends the Jenkins et al paper. In addition, there is undeniably a lot of interest in the biology of Rab11. However a number of aspects of the work need addition before it is suitable for publication.

a) The analysis of point mutations in SH3BP5 and Rab11 is a useful addition, although the value of some of the Rab11 mutations is perhaps questionable. Some of the Rab11 mutations are in the switch I and switch II regions, in residues that are conserved in many Rabs, and so they could have an effect on nucleotide binding and/or Mg²⁺ coordination, (particularly those in switch I). This might explain why they cannot fit the GTP exchange behaviour of a few of these mutants to an exponential function (Supplemental Figure 4) or why, in Figure 4, Rab11a K41A behaves so differently to the wild-type and the other mutants in the presence of EDTA. This needs to be discussed properly, and consideration given to removing mutants that are not informative.

b) The authors show that only alpha-helices 1 and 4 are needed for Rab11a activation and they point out the flexibility of the hinge between $\alpha 2$ and $\alpha 3$, (in the Rab11 bound crystal, residues inside this hinge adopt a completely different conformation). The authors suggest that this phenomenon "might have some switching functions independent of the GEF reaction". What functions do they have in mind? - the authors need to clarify this point.

c) To test Rab11 activation using in vitro GEF assays the authors use two different versions of Rab11a, Rab11a (1-173) and Rab11a (1-211), but they don't explain which one they are using although they found Rab11a (1-173) is more sensitive to EDTA.

d) Page 9: "However, this hydrogen bond is dispensable for the GEF activity because the Q63A mutation of SH3BP5 showed little effect on the GEF activity (Fig. 3C, D and Supplementary Fig. 4C)." Q63A is not a SH3BP5 residue, but it belongs to the Rab11a sequence.

On this topic, the authors say that Q63 forms a hydrogen bond with Y243, but this hydrogen bond is dispensable since the Q63A has just a "little effect on the GEF activity". In figure 3, section C, they show clearly this hydrogen bond between Q63 and Y243 and, in section D, we can see how the mutant Y243A completely prevents the SH3BP5 GEFs activity. How can the authors explain this discrepancy? Is Y243 interacting with another residue or it is just the Y243A mutations affecting the folding of SH3BP5? Experiment to check the protein stability of the different mutants would be helpful.

e) Page 10. It is difficult to calculate a k_{obs} with the time resolution they are using in the GEFs assays (due to the use of a plate reader), nevertheless it is hard to believe, looking at the plots shown in Figure 4, that the mutant E39A reduces the GEF activity by 50%, while I44A does it by 25%. It seems from the graphs that E39A has no effect, while I44A abolishes the GEF activity. If is the case, it again raises a discrepancy between the effect on a residue in Rab11 and its putative interacting residue on SH3BP5, H258 this time. H258A abolishes SH3BP5 GEF activity.

f) In Figure 5, the authors appear to use a different method to measure the GTP exchange rate (different time frame, % fluorescence...). They want to show that SH3BP5 and SH3BP5L GEF activity is specific, but this method is not described in Materials and Methods section. Surprisingly, what it is mentioned in the Methods is that they have done the experiment shown in Figure 5, Section I in the presence of liposomes, and they provide the liposome composition. However, the effect of liposomes in this experiment is not mentioned in the results section nor shown in Figure 5.

g) At several places in the text, they refer to Jenkins et al 2018 as the "preceding paper" (mainly in the Discussion), which is a bit confusing.

h) The last part of the Discussion is difficult to follow since some of the structures they refer to are not depicted in Supplemental figure 7 (e.g. Rab21-Vps9), and they describe the mechanism of Rab activation by different GEFs, but without mentioning which Rab is a substrate of these GEFs.

i) Minor issues:

Abstract: "SH3BP5 and SH3BP5-like proteins have recently been found to serve as a guanine nucleotide exchange factor (GEF) for Rab11". Needs grammatical correction.

Page 3: "Rab is reversibly anchored to membranes by the posttranslational geranylgeranylation of the C-terminal cysteine residues". Misspelling

Figure1/Figure1 legend. Sections B, C, and D in the legend do not correspond to these panels in the Figure.

Figure 2: Mg²⁺ is not depicted in Panel B, Rab11a-GDP

Page 14: Protein preparation: The methods used to culture the cells are not described.

Please improve the colors in the GEF assay plots: grey traces are difficult to differentiate.

Reviewer #1 (Comments to the Authors (Required)):

The authors have determined by x-ray crystallography a structure of SH3BP5, a Rab11 GEF with a novel fold and activation mechanism, both by itself and in complex with Rab11. Although modest, their resolution is sufficient for them to elucidate the mechanism for activation. Their proposed mechanism is consistent with their thorough mutational/functional analysis. They also determine the basis for Rab11 selectivity.

The results represent a significant advance in the field Rab regulation and are confirmatory/complimentary of independent work published just slightly earlier (in Sept 2018) by the JE Burke group. The authors of the present manuscript have a more extended study in that they have also determined the structure of SH3BP5 alone, although this is not all that informative and does not add much in terms of mechanistic insight, and they present some localization data.

I strongly support publication of this well-done work by LSA.

We appreciate his/her favorable comment.

Reviewer #2 (Comments to the Authors (Required)):

This is a generally well performed study of SH3BP5 as an exchange factor (GEF) for Rab11. The structural results obtained show that the GEF domain of SH3BP5 forms a v-shaped coiled coil structure, but its complex with Rab11 has a completely different overall topology to that of other GEF:GEF complexes that are coiled coil structures (Sec2 or Rabin8 in complex Sec4 or Rab8, respectively). Nevertheless, and even if the authors go to some lengths to point out the supposedly unique characteristics of this particular GTPase:GEF mechanism, it still adheres to the main principles seen (disturbance of Switch I and Switch II structures, displacement of Mg²⁺ ions) in other cases. The difference is in the manner in which these changes are induced.

... When discussing the mechanisms of other Rab GEFs, the authors emphasize that in one case, the loss of an interaction of the guanine base with a phenylalanine at the start of the Switch I region is important. However, this is probably true of all Ras family GTPase-GEF interactions, including that described here, and should not be emphasized as an individual feature of one mechanism. Looking at the present structure from the coordinates (from pdb code 6djl), it can

be seen that Phe36 is far removed from where the guanine base would be if GDP were bound. This can be nicely seen if the structure is superimposed on that of Rab8 in the Rab8:GDP:Rabin8 structure (pdb code 4lhy).

In page 13, the description of the GEF mechanism by DENN domain was changed as follows:

DENND1B, a member of the DENN domain GEF family, opens the nucleotide-binding pocket of Rab35 by displacing the switch I region, and deforms the switch II region to avoid the steric clash between Rab35 and DENN1B (Supplementary Figure 7C).

The results in the manuscript agree essentially with those in an earlier paper by Jenkins et al., published on-line in Nat. Comm. On September 14, 2018. This paper is properly referred to in the manuscript. There are no significant differences in the reported results and interpretations. Common to both papers is a rather poorly resolved structure (3.8 Å in this contribution, 3.1 Å in the already published paper). The relatively poor resolution is perhaps the reason for delaying publication, i.e. the authors were probably still trying to improve their crystals, and have now resigned themselves to having to make do with the poor resolution of their structure (even worse than in the published work). Presumably, all interpretations made could be done on the basis of the electron density map from the authors' data, but they did have access to the coordinates of the "competition", at least from September on (pdb code 6djl deposited on 2018-05-25, released on 2018-09-26).

Below are a few minor points:

The fluorescence assay should not be referred to as "our" assay (in the abstract). This is a standard assay that has been used very frequently since the early 90s of the last century, with minor variations. In the present case, I wonder why the authors used GppNHp to displace mantGDP? This is quite expensive, and due to the symmetrical manner in which GEFs function, GDP could just as well used (it is the MantGDP release that is rate limiting, not the GTP or GDP binding, as long as they are present at high concentration). GDP is much cheaper than GppNHp, and GTP itself is even cheaper. Why not use GTP? The hydrolysis reaction is of no consequence.

“Our” was removed from the sentences describing fluorescence-based assays.

In our experience, GEF assays using mant-nucleotides stably work in most cases, while GEF assays based on tryptophan fluorescence show lower signals and do not work well in some cases. We therefore favor the assay using mant-nucleotides. We also favor the usage of GppNHp to avoid hydrolysis of GTP by intrinsic GTPase activity of small GTPases, although the activity is generally low and the hydrolysis is of little consequence, as he/she pointed out.

Another point here: the authors used MantGDP from Molecular Probes, which is a mixture of 2 isomers. It is well known that these can have different properties, although that is probably not of importance for the semi-quantitative analysis performed here. Still, it would be better in general to use 3'-mant-2'-deoxy GDP, which does not have this problem. As far as I can see, the "competition" did use this, but referred to it as MantGDP (should have been mantdGDP). See below for a further point relevant to these arguments.

We appreciate this suggestion and will consider the usage of 3'-mant-2'-deoxy GDP in future assays.

In the Introduction, second paragraph: Mg²⁺ is not involved in hydrogen bonds (and cannot be).

We corrected the description in page 3 as follows:

“Switch I and switch II adopt different conformations, depending on the nucleotide-bound state.”

In Fig. 4a, "no enzyme" should read "no GEF", since GEF is not, strictly speaking, an enzyme.

“no enzyme” was changed to “no GEF” in Supplementary Figs. 3C and 4A.

Also, in this figure, and in Fig. 3, why is there a relatively rapid phase at the beginning of the dissociation of MantGDP without added GEF? This could, conceivably, be due to the difference in binding affinities of the 2 isomers alluded to above. In fact, for HRas, Rensland et al. (Biochemistry 30, 11181-11185 (1991); see p. 1183, right hand column) concluded that the 2' isomer bound almost a factor of 10 more weakly than the 3'-isomer, and that this difference was

due to an increased off-rate. A test with 3'-mant-2'-deoxyGDP could resolve this question.

On the basis of this comment, the following explanation was added in page 9.

“A rapidly decreasing phase was observed in the beginning of the curves without GEF. Mant-GDP used in this study was a mixture of 2'-Mant-GDP and 3'-Mant-GDP. 2'-Mant-3'-deoxy-GDP reportedly has 10-times faster dissociation rate than 3'-Mant-2'-deoxy-GDP [22]. The rapidly decreasing phase might reflect the faster release of 2'-Mant-GDP.”

Reviewer #3 (Comments to the Authors (Required)):

SH3BP5 is an exchange factor for the small GTPase Rab11 that acts in membrane traffic on recycling endosomes and the trans-Golgi network. Sakurako et al. describe the crystal structure of the GEF domain of SH3BP5 alone and in a complex with Rab11a. The authors find that SH3BP5 folds into a V shape coiled-coil architecture, and its interaction with Rab11a results in a rearrangement of the Rab switch I region. The rearrangement of switch I releases the GDP bound to the GTPase and so facilitates nucleotide exchange.

This structural analysis is of a good quality and convincing, and it is consistent with a structure for SH3BP5 in complex with Rab11 that was published a few months ago (Jenkins et al (2018) Nat Comms 9, 3772). In addition, the authors of this manuscript extend their work beyond that of Jenkins et al, by performing an extensive analysis of different mutants forms of both proteins, Rab11 and SH3BP5, in GEF assays. Overall this is a valuable addition to the field and it both confirms and extends the Jenkins et al paper. In addition, there is undeniably a lot of interest in the biology of Rab11. However a number of aspects of the work need addition before it is suitable for publication.

a) The analysis of point mutations in SH3BP5 and Rab11 is a useful addition, although the value of some of the Rab11 mutations is perhaps questionable. Some of the Rab11 mutations are in the switch I and switch II regions, in residues that are conserved in many Rabs, and so they could have an effect on nucleotide binding and/or Mg+2 coordination, (particularly those in switch I). This might explain why they cannot fit the GTP exchange behaviour of a few of these mutants to an exponential function (Supplemental Figure 4) or why, in Figure 4, Rab11a K41A behaves so differently to the wild-type and the other mutants in the presence of EDTA. This needs to be discussed properly, and consideration given to removing mutants that are not

informative.

Among the switch-I and switch-II mutation sites examined in this study, only Thr43 was obviously involved in the nucleotide binding through the coordination of Mg²⁺. We believe that all tested mutants are informative, and added the following explanation in page 10:

“However, it should be noted that the side chain of Thr43 coordinates Mg²⁺ in the nucleotide-bound Rab11a. The T43A mutation might affect the nucleotide binding, although Mant-GDP could be loaded to the Rab11a T43A mutant as efficiently as to wild type.”

On the other hand, the structure provides no clues to explain why the Rab11A K41A mutant is sensitive to EDTA in the context of the nucleotide release.

b) The authors show that only alpha-helices 1 and 4 are needed for Rab11a activation and they point out the flexibility of the hinge between $\alpha 2$ and $\alpha 3$, (in the Rab11 bound crystal, residues inside this hinge adopt a completely different conformation). The authors suggest that this phenomenon "might have some switching functions independent of the GEF reaction". What functions do they have in mind? - the authors need to clarify this point.

We added the following description in page 8:

“This flexibility of the edge of the $\alpha 2/\alpha 3$ coiled coil, which is located far from the GEF catalytic site, might have some switching functions independent of the GEF reaction, such as binding to other proteins or membrane lipids.”

We also mentioned this point in the discussion (page 14) as follows:

“... the edge of the $\alpha 2/\alpha 3$ coiled coil ... might play some switching roles independent of the GEF reaction, possibly through the binding to other proteins and/or membrane lipids.”

c) To test Rab11 activation using in vitro GEF assays the authors use two different versions of Rab11a, Rab11a (1-173) and Rab11a (1-211), but they don't explain which one they are using although they found Rab11a (1-173) is more sensitive to EDTA.

This point was mentioned in page 8 as follows:

“Rab11a (1–173) and Rab11a (1–211) were used for the assays of SH3BP5 and Rab11a mutants, respectively.”

d) Page 9: "However, this hydrogen bond is dispensable for the GEF activity because the Q63A mutation of SH3BP5 showed little effect on the GEF activity (Fig. 3C, D and Supplementary Fig. 4C)." Q63A is not a SH3BP5 residue, but it belongs to the Rab11a sequence.

“... the Q63A mutation of SH3BP5 ...” was corrected to “... the Q63A mutation of Rab11a...” in page 9.

On this topic, the authors say that Q63 forms a hydrogen bond with Y243, but this hydrogen bond is dispensable since the Q63A has just a "little effect on the GEF activity". In figure 3, section C, they show clearly this hydrogen bond between Q63 and Y243 and, in section D, we can see how the mutant Y243A completely prevents the SH3BP5 GEFs activity. How can the authors explain this discrepancy? Is Y243 interacting with another residue or it is just the Y243A mutations affecting the folding of SH3BP5? Experiment to check the protein stability of the different mutants would be helpful.

We added the following explanation regarding the function of Tyr243 in page 10:

“Therefore, the decrease in GEF activity of SH3BP5 Y243A mutant mainly depends on the loss of hydrophobic interaction.”

e) Page 10. It is difficult to calculate a k_{obs} with the time resolution they are using in the GEFs assays (due to the use of a plate reader), nevertheless it is hard to believe, looking at the plots shown in Figure 4, that the mutant E39A reduces the GEF activity by 50%, while I44A does it by 25%. It seems from the graphs that E39A has no effect, while I44A abolishes the GEF activity. If is the case, it again raises a discrepancy between the effect on a residue in Rab11 and its putative interacting residue on SH3BP5, H258 this time. H258A abolishes SH3BP5 GEF activity.

There is a short time lag between the starting points of the reaction and measurement in the assay using a standard plate reader. We made our best efforts to minimize and

uniform such time lag in the GEF assay. Indeed, three independent measurements showed similar k_{obs} values for wild-type or E39A Rab11a as shown in the table below.

k_{obs} (s^{-1}) \times $\times 10^{-3}$	Assay 1	Assay 2	Assay 3
Wild type	31.5	35.3	30.9
E39A	19.9	20.6	17.2

We believe that the k_{obs} values estimated from our mutational experiments should reflect the effect of the individual mutations. Plate readers have actually been used for GEF assays (*e.g.*, Delprato *et al.*, “Structure, exchange determinants, and family-wide rab specificity of the tandem helical bundle and Vps9 domains of Rabex-5.” *Cell*, **118**, 607-17 (2004)).

The I44A mutant of Rab11a eliminates the GEF activity as described in page 10:

“On the other hand, the T43A and I44A mutations eliminated the activity.”

As for SH3BP5 H258A, the side chain of SH3BP5 His258 forms a hydrogen bond with the main-chain CO group of Rab11a Glu39. This explains why Rab11a E39A retains the activity while SH3BP5 H258A drastically reduces the activity.”

f) In Figure 5, the authors appear to use a different method to measure the GTP exchange rate (different time frame, % fluorescence...). They want to show that SH3BP5 and SH3BP5L GEF activity is specific, but this method is not described in Materials and Methods section. Surprisingly, what it is mentioned in the Methods is that they have done the experiment shown in Figure 5, Section 1 in the presence of liposomes, and they provide the liposome composition. However, the effect of liposomes in this experiment is not mentioned in the results section nor shown in Figure 5.

The method for the GEF assay of SH3BP5 and SHBP5L on liposomes was added to the Materials and Methods section (page 18-19).

The GEF reaction in solution and that on membrane cannot be simply compared because their reaction conditions and environments are different. Therefore, we will not mention the effect of liposome for the GEF assay in this paper.

g) At several places in the text, they refer to Jenkins et al 2018 as the "preceding paper" (mainly in the Discussion), which is a bit confusing.

We removed the phrase "preceding paper".

h) The last part of the Discussion is difficult to follow since some of the structures they refer to are not depicted in Supplemental figure 7 (e.g. Rab21-Vps9), and they describe the mechanism of Rab activation by different GEFs, but without mentioning which Rab is a substrate of these GEFs.

The structure of the Rab21–Rabex5 Vps9 complex was additionally shown in Supplementary Figure 7. Each complex was labeled as A-G. The characteristic feature of each complex was described in the main text with the reference related to each panel. The substrate Rab for each GEF was described in the text.

i) Minor issues:

Abstract: "SH3BP5 and SH3BP5-like proteins have recently been found to serve as a guanine nucleotide exchange factor (GEF) for Rab11". Needs grammatical correction.

This sentence was corrected as follows:

“SH3BP5 and SH3BP5-like (SH3BP5L) proteins have recently been found to serve as guanine nucleotide exchange factors (GEFs) for Rab11.”

Page 3: "Rab is reversibly anchored to membranes by the posttranslational geranylgeranylation of the C-terminal cysteine residues". Misspelling

“geranylgeranylation” was corrected to “geranylgeranylation”.

Figure1/Figure1 legend. Sections B, C, and D in the legend do not correspond to these panels in the Figure.

They were arranged in the right order.

Figure 2: Mg²⁺ is not depicted in Panel B, Rab11a-GDP

The Rab11a-GDP structure in Figure 2B is derived from PDB 1OIV, which contains no Mg²⁺.

Page 14: Protein preparation: The methods used to culture the cells are not described.

The methods for the cell culture were added in page 15 as follows:

“The cells were cultured at 37 °C in LB medium supplemented with 50 mg/L ampicillin. The protein expression was induced by 0.2 mM IPTG when A₆₀₀ reached 0.6-0.8, and the culture was continued overnight at 15 °C. The SeMet-labeled His₆-SUMO-tagged SH3BP5-RA/M167A was expressed in *E. coli* B834 strain. The cells were cultured at 37 °C in the customized medium equivalent to LeMaster medium (Code No. 06780, Nacalai tesque) with 50 mg/L SeMet. After the induction by 0.2 mM IPTG, the cells were further grown overnight at 20 °C”.

Please improve the colors in the GEF assay plots: grey traces are difficult to differentiate.

The colors of the curves in the GEF assay were changed in Figures 3 and 4, and Supplementary Figures 3 and 4.

March 5, 2019

RE: Life Science Alliance Manuscript #LSA-2019-00297-TR

Dr. Shuya Fukai
Institute of Molecular and Cellular Biosciences, University of Tokyo
Life Science Division, Synchrotron Radiation Research Organization
1-1-1 Yayoi, Bunkyo-ku
306 Main Building
Tokyo 113-0032
Japan

Dear Dr. Fukai,

Thank you for submitting your revised manuscript entitled "Structural basis of guanine nucleotide exchange for Rab11 by SH3BP5". We appreciate the introduced changes and would thus be happy to publish your paper in Life Science Alliance pending final revisions necessary to meet our formatting guidelines:

- please add a callout in the manuscript text to Fig4D and S5A
- please move the supplementary figure legends into the main manuscript text file and upload the S figures as separate, individual files
- please indicate where missing (eg GEF assays) how many replicates of the experiments shown have been performed
- please note that figure legend 1 mentions panel (C) twice at the moment (instead of (D))
- please link your profile in our submission system to your ORCID iD, you should have received an email with instructions on how to do so

A. FINAL FILES:

- An editable version of the final text (.DOC or .DOCX) is needed for copyediting (no PDFs).
- High-resolution figure, supplementary figure and video files uploaded as individual files: See our detailed guidelines for preparing your production-ready images, <http://www.life-science-alliance.org/authors>
- Summary blurb (enter in submission system): A short text summarizing in a single sentence the

study (max. 200 characters including spaces). This text is used in conjunction with the titles of papers, hence should be informative and complementary to the title. It should describe the context and significance of the findings for a general readership; it should be written in the present tense and refer to the work in the third person. Author names should not be mentioned.

B. MANUSCRIPT ORGANIZATION AND FORMATTING:

Sincerely,

Andrea Leibfried, PhD
Executive Editor
Life Science Alliance
Meyrhofstr. 1
69117 Heidelberg, Germany
t +49 6221 8891 502
e a.leibfried@life-science-alliance.org
www.life-science-alliance.org

March 8, 2019

RE: Life Science Alliance Manuscript #LSA-2019-00297-TRR

Dr. Shuya Fukai
Institute of Molecular and Cellular Biosciences, University of Tokyo
Life Science Division, Synchrotron Radiation Research Organization
1-1-1 Yayoi, Bunkyo-ku
306 IQB Main Building
Tokyo 113-0032
Japan

Dear Dr. Fukai,

Thank you for submitting your Research Article entitled "Structural basis of guanine nucleotide exchange for Rab11 by SH3BP5". It is a pleasure to let you know that your manuscript is now accepted for publication in Life Science Alliance. Congratulations on this interesting work.

DISTRIBUTION OF MATERIALS:

Again, congratulations on a very nice paper. I hope you found the review process to be constructive and are pleased with how the manuscript was handled editorially. We look forward to future exciting submissions from your lab.

Sincerely,
